# Objective Classification of Glistening in Implanted Intraocular Lenses Using Optical Coherence Tomography: Proposal for a New Classification and Grading System

**DOI:** 10.3390/jcm12062351

**Published:** 2023-03-17

**Authors:** José Ignacio Fernández-Vigo, Bárbara Burgos-Blasco, Lucía De-Pablo-Gómez-de-Liaño, Inés Sánchez-Guillén, Virginia Albitre-Barca, Susana Fernández-Aragón, José Ángel Fernández-Vigo, Ana Macarro-Merino

**Affiliations:** 1Centro Internacional de Oftalmología Avanzada, 28010 Madrid, Spain; 2Department of Ophthalmology, Hospital Clínico San Carlos, Instituto de Investigación Sanitaria (IdISSC), 28040 Madrid, Spain; 3Department of Ophthalmology, Hospital Universitario 12 de Octubre, 28041 Madrid, Spain; 4Department of Ophthalmology, Hospital Perpetuo Socorro, 06010 Badajoz, Spain; 5Centro Internacional de Oftalmología Avanzada, 06011 Badajoz, Spain; 6Department of Ophthalmology, Universidad de Extremadura, 06006 Badajoz, Spain

**Keywords:** glistening, intraocular lens, opacification, optical coherence tomography

## Abstract

Purpose: To propose a classification of the glistening in intraocular lenses (IOL) using swept-source optical coherence tomography (SS-OCT) by means of a simple, objective and reproducible method that allows the quantification of the presence and severity of glistening. Methods: A cross-sectional study on a sample of 150 eyes of 150 patients who underwent cataract surgery in at least 600 days before the exam and attended a routine examination. Each subject was examined by SS-OCT after pupil dilation, identifying the presence of glistening or hyperreflective foci (HRF) in the central area of the IOL. The degree of glistening was classified into four categories: 0: ≤5 HRF; 1: 6 to 15 HRF; 2: 16 to 30 HRF; and 3: >30 HRF. The intra and interobserver reproducibility (intraclass correlation coefficient, ICC) in the quantification and classification of the glistening were calculated. The correlation between the horizontal and vertical scan of the IOL was also assessed. Results: Glistening was present in the IOL in 42.7% of the patients. The mean number of HRF or glistening microvacuoles was 10.4 ± 26.2 (range 0 to 239). In total, 63.3% of the IOLs had a grade 0, 20% grade a 1, 6.7% grade a 2 and 10% a grade 3. The intraobserver and interobserver reproducibility were very high, both for the absolute quantification of the glistening (ICC ≥ 0.994) and for the severity scale (ICC ≥ 0.967). There was an excellent correlation in the quantification of the IOL glistening between the horizontal and vertical scans (R ≥ 0.834; *p* < 0.001). Conclusions: The use of SS-OCT makes it possible to identify, quantify and classify IOL glistening in a simple, objective and reproducible way. This technique could provide relevant information for the study of the glistening on IOLs.

## 1. Introduction

Cataract surgery with an intraocular lens (IOL) implant is one of the most frequent surgeries performed worldwide [1]. There are two major complications specifically related to IOL. The first is serious but infrequent and consists in the opacification of the IOL, while the second, which is the glistening of the lens, is more frequent but generally described as less important [2,3,4,5].

Intraocular lens glistening is the presence of small (1 to 20 µm), shiny, white or yellow spots, which are fluid-filled microvacuoles (MVs), within the IOL after its implantation [2,6]. In the proposed mechanism in its formation, the IOL polymer absorbs water, which forms MV with the IOL material. The difference in the refractive index between the water and the IOL polymer results in their characteristic appearance on the slit-lamp exam [7]. According to various authors, IOL glistening is influenced by the manufacturing process, the packaging system, the changes in temperature, the equilibrium water content, the IOL model, the IOL power, the breakdown of the blood–retina or blood–water barrier and the postoperative inflammation, especially in combined surgeries [3,6,8,9]. This glistening, or hydration-related phenomenon, has been observed in a variety of materials, including silicone, hydrogel and poly methyl methacrylate (PMMA) IOLs, but it is particularly common in hydrophobic acrylic IOLs [1,6,10,11].

The symptoms of glistening may include a decrease in vision, halos or glare, or a decrease in contrast sensitivity, although most studies cite no apparent effect of IOL glistening on visual acuity [2,6,12]. For these reasons, along with others, IOL explant has rarely been reported in the literature on IOL glistening [13]. However, the full impact of IOL glistening on postoperative visual function and its changes in the long term after surgery remain to be fully elucidated.

The first observations of this phenomenon of glistening formation on IOL materials were performed in vitro and in clinical practice after slit-lamp examinations [7,14,15,16,17,18]. However, the latter presents two main difficulties: it is a time-consuming technique and great photographic skill is required by an expert examiner to obtain valuable images of the glistening. Furthermore, this process usually requires the post-processing of the images. For these reasons, most authors detect IOL glistening using a slit-lamp examination and quantify them subjectively, based on a rating scale. Currently, one of the most popular glistening-classification systems is the Miyata scale, which is used to classify glistening in IOL, ranging from grade 0 (no glistening) to grade 3 (severe glistening) [19]. The lack of an objective, examiner-independent and reproducible method for in vivo glistening evaluation is a common problem [8].

Recently, different authors proposed the use of Scheimpflug-camera-based devices to assess and classify the glistening in IOLs. However, this technology does not provide images with sufficient resolution to perform an automated glistening count and classification. Optical coherence tomography (OCT) is a noninvasive imaging technique that has revolutionized ophthalmological diagnoses in several areas and could be a helpful tool in this area.

To date, no objective, fast and reproducible method for the assessment of IOL glistening in clinical practice has been described. In addition, no studies have focused on the utility of OCT in the assessment of IOL glistening in vivo, and only one clinical study has reported an OCT analysis of IOL glistening [20].

Therefore, the main objectives of the present study are to assess the utility of the OCT to identify and classify the severity of the glistening, as well as assessing the reproducibility of this classification system.

## 2. Methods

### 2.1. Patients

A cross-sectional study was conducted on 150 patients recruited consecutively from those attending the Centro Internacional de Oftalmologia Avanzada (Madrid, Spain) for a routine ophthalmological examination over the period of 1 November 2022, to 1 December 2022.

Subjects were invited to participate if they met all the inclusion criteria and none of the exclusion criteria after giving their written informed consent. The study protocol adhered to the tenets of the Declaration of Helsinki and was approved by the Center’s Review Board.

Inclusion criteria were patients who underwent cataract surgery at least 600 days before the recruitment process began, with the SN60WF IOL models made of Acrysof hydrophobic acrylic material (Alcon, Fort Worth, TX, USA), and who were consulted for a routine examination. Exclusion criteria were complications during cataract surgery and postoperative uveitis. Furthermore, images of insufficient quality, determined as a signal strength intensity (SSI) ≤ 2, or with artefacts, were excluded.

One eye from each patient was randomly included in the general study, while both eyes from the same patient were included in the reproducibility study.

### 2.2. Study Protocol

The subjects enrolled provided their medical history and underwent a complete ophthalmological examination, including anterior-segment OCT (SS-OCT). The ophthalmological exam included visual acuity and cycloplegic refractive error, slit-lamp biomicroscopy, tonometry with Canon TX 10^®^ pneumotonometer (Canon Inc., Tokyo, Japan) and posterior-segment ophthalmoscopy. The IOL model and the time since the surgery were registered. All the examinations were performed on the same day.

For the assessment of the IOL, the SS-OCT device employed was a DRI-Triton^®^ (Topcon Corporation, Tokyo, Japan), which uses a central wavelength of 1050 nm with an axial resolution of 8 μm, a transverse resolution of 20 μm and a scanning speed of 100,000 A-scans per second. To obtain cross-sectional images of the IOL, we employed the anterior-segment lens of the device using the “line” anterior-segment capture mode of a 6 mm exploration field on the horizontal axis. Furthermore, a radial scan, which consisted in 12 scans centered on the pupil, was conducted. This type of scan was intentionally performed to study the vertical scan of the IOL in order to analyze the correlation between the severity of the glistening and the horizontal scan. All OCT images were acquired after pupil dilation by a well-trained examiner (JIFV), with the subjects sitting up. Only images of sufficient quality, defined as a signal-strength intensity (SSI) above 3, were accepted.

### 2.3. Assessment of the IOL Glistening by OCT

The presence of glistening was identified as hyperreflective foci (HRF) or MVs inside the optic of the IOL, between the anterior surface and posterior surface of the lens (Figure 1).

Based on a pilot study, the degree of glistening was classified into four categories: 0: ≤5 HRF; 1: 6 to 15 HRF; 2: 16 to 30 HRF; and 3: >30 HRF. Therefore, the glistening was described as the absolute number of HRF and as the severity grading (Figure 2). The evaluation of the OCT images was performed by two experienced examiners (JIFV and JAFV).

In a subgroup of 50 eyes from 25 patients, which were randomly selected, two OCT scans of the IOL were performed on the same day, separated by an interval of 2 min, to assess the repeatability of the images. It should be noted that no eye-tracking on the IOL was applied because it is not an available option in the device’s software.

For intra and interobserver reproducibility, a different group of 50 eyes from 25 patients was assessed and examined once. Measurements were masked and independently conducted on the images by two observers (JIFV and JAFV) for interobserver reproducibility. To determine intraobserver reproducibility, one of the observers (JAFV) also took measurements of the same images one week after the first measurements.

### 2.4. Statistical Analysis

All statistical tests were performed using the software package, SPSS^®^ (Statistical Package for Social Sciences, v25.0; SPSS Inc., Chicago, IL, USA). Quantitative data are provided as the mean and standard deviation. Qualitative data are expressed as their frequency distributions. Correlations were assessed through Pearson’s correlation coefficients. In the reproducibility analysis, for each measurement, the intraclass correlation coefficient (ICC; two-way mixed effects, absolute agreement, single measurement) was calculated for the two consecutive scans or measurements. Significance was set at *p* < 0.05.

## 3. Results

Demographics of the 150 patients included in the present study are presented in Table 1.

Intraocular lens glistening was detected in 42.7% of the patients included. The mean number of HRF or glistening dots was 10.4 ± 26.2, with a range from 0 to 239.

Regarding the severity of the glistening in the IOL, 63.3% of the eyes (95/150) had a grade of 0, 20% (30/150) had a grade of 1, 6.7% (10/150) had a grade of 2 and 10% (15/150) had a grade of 3. The numbers of MVs or HRF were as follows: in grade 0, 1.0 ± 1.4 (0 to 5); in grade 1, 9.6 ± 2.5 (6 to 15); in grade 2, 20.7 ± 4.6 (16 to 30); and in grade 3, 74.5 ± 46.2 (31 to 239).

The intraobserver and interobserver reproducibility were very high, both for the absolute quantification of the glistening (ICC ≥ 0.994) and for the severity scale (ICC ≥ 0.967). The repeatability for the absolute quantification and for the severity was also very high (ICC ≥ 0.955; Table 2).

There was an excellent correlation in the quantification of the IOL glistening between the horizontal and vertical scan, for both the right and the left eyes (R = 0.921 and R = 0.834 respectively; *p* < 0.001; Figure 3). However, the correlation in the vertical scans between the right and left eyes was moderate (R = 0.566; *p* < 0.001), and very similar to that in the horizontal scans between the right and left eyes (R = 0.582; *p* < 0.001).

There was a moderate correlation between the time since surgery and the severity of the glistening (R = 0.497; *p* < 0.0001). In addition, there was a weak correlation between age and the degree of glistening (R = 0.174; *p* = 0.034; Figure 3).

## 4. Discussion

Optical clarity is an important parameter in the quality assessment of implanted IOLs after cataract surgery [6]. Among different causes of IOL opacifications, glistening is relatively frequent [5]. Glistening or small fluid-filled vacuoles in the IOL material appear in the form of refractive particles that glisten upon slit-lamp examination due to hydration-related phenomena in hydrophobic acrylic lenses when the latter are in aqueous environments.

In the present study, SS-OCT was used to identify, quantify and classify IOL glistening in a simple and objective way. We analyzed the presence of glistening along the entire IOL optic shown on the OCT scans, detecting the absolute number of MVs, seen as HRF, and proposing an OCT-based classification. This classification system presented excellent reproducibility and a very strong correlation between the horizontal and vertical scans. Therefore, the use of only one OCT scan makes it possible to classify the degree of glistening, without the need for a cube or raster scan. Another advantage is that the entire optic in an OCT scan is assessed; it is not limited to a small area of 1 mm^2^.

Werner et al. were the first authors to observe glistening through anterior-segment OCT (AS-OCT), by analyzing ex vivo hydrophobic acrylic IOLs explanted because of various complications [21]. They stated that this technique was helpful in analyzing the location and density of the glistening, and AS-OCT scans showed that the glistening was homogeneously distributed within the entire optic, although it was sometimes absent in a small subsurface area. Subsequently, in 2019, Tripathy and Sridhar reported an image describing IOL glistening through OCT in one patient [20]. Glistening takes the form of hyperreflective dots inside the optic of a lens. By contrast, pigment deposits or pseudoexfoliative materials are deposits over the surface of the IOL [21,22].

Currently, one of the most popular glistening-classification systems is the Miyata scale, which is based on slit-lamp images and features the following grades: grade 0, no glistening; grade 1, up to 50/mm^2^; grade 2, up to 100/mm^2^; and grade 3, 200/mm^2^ (considered severe glistening) [19]. Colin et al., who also used slit-lamp images, graded and quantified glistening in a central area measuring 0.75 to 2 mm using Image J, applying different processing techniques. The size limits were set so that the Image J program would recognize MVs as features with sizes between 0 and 0.001 mm^2^. The mean values for the objective grading of the glistening were 26 ± 43 MVs/mm^2^ for grade 0, 152 ± 142 MVs/mm^2^ for grade 1 and 261 ± 139 MVs/mm^2^ for grade 2. Interestingly, in Colin et al.’s study, only four lenses had >400 MVs/mm^2^; the maximum observed density was 597 MVs/mm^2^ [3]. In the present study, the numbers of MVs were as follows: in grade 0, 1.0 ± 1.4; in grade 1, 9.6 ± 2.5; in grade 2, 20.7 ± 4.6; and in grade 3, 74.5 ± 46.2. The highest number observed was 239.

Similarly, in microscopic images obtained with a digital camera, Yildirim et al. assessed the number of MVs in the central area and four peripheral IOL sections [23]. The software automatically calculated the number of MVs and a modification of the Miyata scale was established: grade 0 (<25 MVs/mm^2^); grade 1 (25–100 MVs/mm^2^); grade 2 (100–200 MVs/mm^2^); and grade 3 (>200 MVs/mm^2^). The central section was the region with the highest glistening density.

A similar approach to the quantification of glistening was carried out by Stanojcic et al. [24]. The methodology was based on central vertical slit-lamp images of 10 mm × 2 mm at 40°, on which a 1 × 3 mm grid divided into 1 mm^2^ areas was overlayed. The grades of glistening density were assigned according to an 8-point ordinal scale based on increments of 10 glistenings/mm^2^ from 0 (grade 0) up to more than 60 (grade 7) [24,25]. Other study groups classified glistening based on MV size (6–25 µm and over 25 µm) [26]. In the present study, based on the OCT resolution, both size groups were detected by using this technology.

In the last few years, a new proposal to classify glistening based on a digital analysis of Scheimpflug images was described by Biwer et al. [8]. They reported a mean glistening count of 2.8 ± 4.8 MV/mm^2^. However, the subjective glistening-grading score showed moderate agreement with a counted score based on slit-lamp images. After this study, the authors acknowledged that the Scheimpflug device did not provide images with sufficient resolution to perform a glistening classification.

To date, different proposals have been developed and employed by different authors to assess and classify the glistening on IOLs. However, a direct comparison of the quantification and classification systems using slit-lamp photography (based on a frontal plane of the IOL) with that proposed here, using OCT (based on a transversal plane of the IOL), is difficult. Most authors quantify glistening in IOLs using an optic area of 1 mm^2^. By contrast, we analyzed the entire optic IOL OCT-scan visualization, which is approximately 3 mm^2^ (6 mm in IOL length and 0.5 mm in IOL thickness). Therefore, this method allows a more comprehensive analysis of IOLs, along with many other advantages.

As Werner et al. recognized, the glistening produced in vitro in their study may result in a morphologic aspect that appears to be stronger on OCT compared with the clinical situation [21]. Therefore, these authors suggested that only clinical studies can confirm whether clinically observed degrees of glistening can be assessed by using AS-OCT. In this regard, the present study demonstrated the ability of this technique to visualize and classify glistening in routine clinical practice.

Interestingly, in the present study, a significant correlation was observed between the time since surgery and the severity of glistening. This was in accordance with the results of a study carried out by Waite et al., who detected an association between the severity index determined by photographs of IOLs and the progression over time [27].

The effect of IOL glistening on visual quality has not yet been thoroughly investigated. The main effect of glistening on vision seems to be an increase in intraocular light scattering and glare, rather than changes in visual acuity [28]. Similarly, Kanclerz et al. found that glistening and subsurface nanoglistening manifest in hydrophobic acrylic IOLs and induce straylight rather than a decrease in visual acuity [1]. A study reported that only MVs greater than 10 µm induce a worsening in the modulation-transfer function [29]. Moreover, Weindler et al. showed that a rather large number of glistenings (>500/ mm^2^) is needed to reduce the optical quality [30]. In this line of study, other classification systems have been proposed. Waite et al. suggested employing a severity index, which was defined as the size of the glistenings multiplied by the density of the glistenings (%area) [27], or analyzing the size as a measure of the area (%area/size) [26,31]. Henriksen et al. concluded that glistening %area, at a key size, was correlated with random light scatter [26].

In recent years, IOL manufacturers have tried to improve the fabrication processes and IOL materials to develop glistening-free hydrophobic acrylic polymers. With the introduction of some materials, such as the popular foldable Acrysof IOL (Alcon Laboratories, Inc.), in which this phenomenon often occurs, IOL glistening received clinical and research attention. Using these IOLs, Thomes and Callaghan induced glistening in vitro and compared AcrySof IOLs manufactured in 2003 and 2012 [17], observing a large decrease from 315.7 MV/mm^2^ to 39.9 MV/mm^2^, respectively [6,32]. Recently, Stanojcic et al. compared two hydrophobic acrylic aspheric monofocal IOLs (Clareon and Tecnis PCB00) in terms of glistening occurrence [25]. At 12 months, the glistening was minimal, with no difference in grade between the groups (*p* = 0.2). The new Clareon material is promising, with a study reporting no glistening up to 9 years after implantation [30]. The amount of fluid in hydrophobic acrylic materials has been shown to be negatively correlated with the occurrence of glistening, so the relatively high water content of the Clareon CNA0T0 material (1.5%) may explain these findings [32,33].

To the best of our knowledge, this is the first study to propose an OCT-based method to identify and classify IOL glistening. The main advantages are that it is a fast and easy exam, that it is highly reproducible and that the image can be directly assessed in vivo, with no need for image-post-processing techniques. By contrast, slit-lamp photographs with a posterior complex requires the processing of the images, which is a time-consuming method and requires an expert examiner. Even when the repeated images were acquired without any eye-tracking, the repeatability of the quantification and of the classification of the glistening on the IOL was excellent. By providing an objective classification, the method allows prospective evaluation and may serve as a valid tool to study new IOL designs and materials for resistance to glistening formation in clinical trials. As previously stated by Werner et al., AS-OCT may be helpful in assessing the presence, location and density of IOL changes, preventing the misdiagnosis of opacification and the performance of unnecessary procedures, such as posterior capsulotomy or explantation [21].

Our study has several limitations. First, establishing a cut-off point to determine the size in which a MV or HRF is considered as glistening was occasionally challenging. The identification of the smallest MV or HRF was limited by the resolution of the OCT, although we obtained excellent reproducibility. There are no strict criteria to determine how many glistening points are allowed in the determination of glistening-free IOLs [25], although, traditionally, most authors have used 50 MVs/mm^2^ as a cutoff [19]. In addition, the density of the MVs or HRF was not considered, so the possible influence of those with a higher density is unknown. In addition, we only studied the central scanning area of the IOLs; although this is the most relevant area, because it includes the visual axis, glistening is usually distributed throughout the entire optic. In the present study, a moderate correlation in the glistening evaluation was observed between both eyes. However, it is well-known that different factors, such as the IOL model and several other factors can affect glistening, and the analysis of inter-eye differences was not the main objective of the present study.

Future studies are warranted to analyze whether glistening progresses over time, as a long follow-up, ideally lasting over 10 years, is needed [24]. Cataract surgery is being performed on increasingly young patients, which, along with the increasing life expectancy, implies that modern IOL needs to maintain optical clarity for several decades [24,25]. The intensity of glistening has been reported to increase for up to 15 years postoperatively, with an increase in surface light scattering on the IOL surface [34]. Studies assessing the possible agreement between OCT-based glistening classification and functional visual tests should be carried out. It would be interesting to establish whether the MV-distribution pattern has clinical relevance. Moreover, future studies should assess the utility of other OCT devices and deep-learning-based algorithms in the provision of automatic descriptions and classifications of glistening; these devices and algorithms may even predict the stability or progression of glistening.

In conclusion, SS-OCT makes it possible to identify, quantify and classify the glistening of IOLs in a simple, objective and reproducible way. This could provide relevant information for the study of the influence of glistening on visual acuity, quality and disability glare.

## Figures and Tables

**Figure 1 jcm-12-02351-f001:**
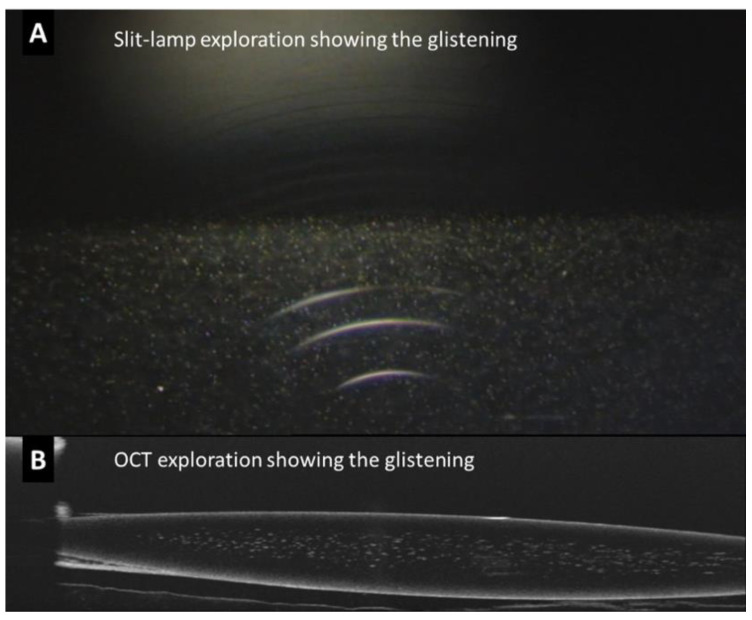
Example of an intraocular lens (IOL) with a high number of hyperreflective foci or microvacuoles in the optic of the lens viewed through slit-lamp photography (**A**) and optical coherence tomography (OCT) (**B**).

**Figure 2 jcm-12-02351-f002:**
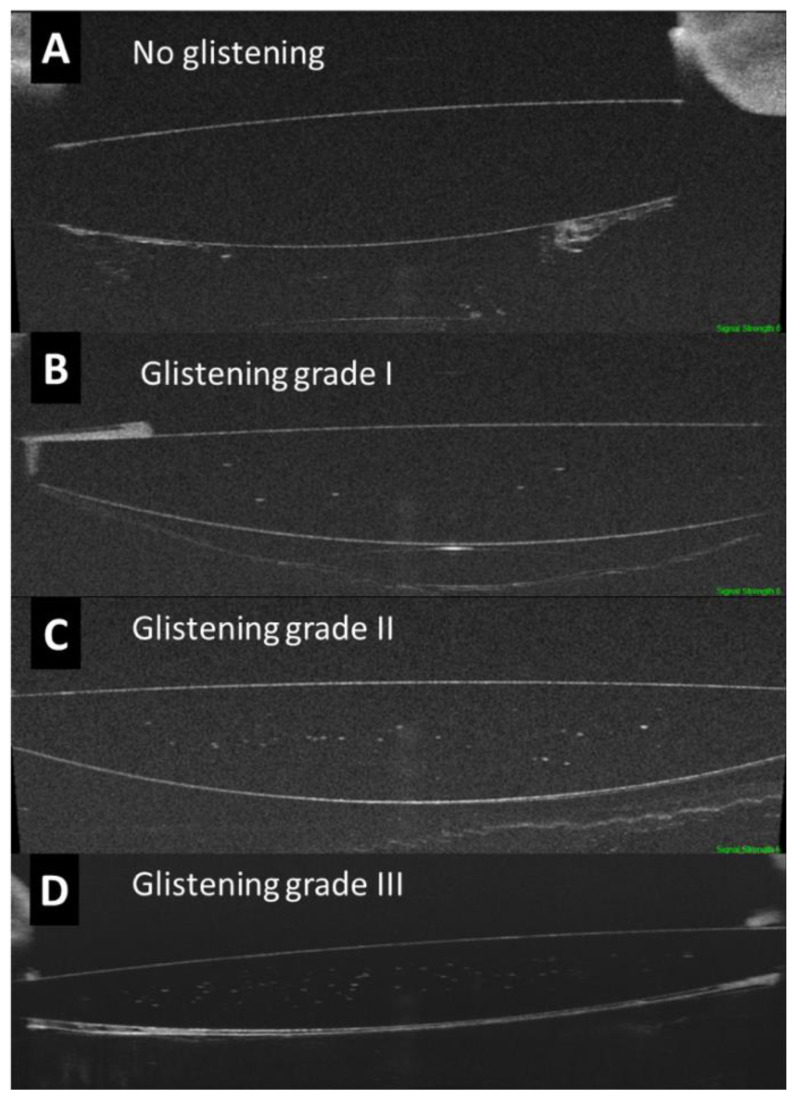
Optical coherence tomography of the optic of the intraocular lens (IOL) assessing the glistening and categorized into 4 groups based on the severity in the amount of hyperreflective foci or microvacuoles.

**Figure 3 jcm-12-02351-f003:**
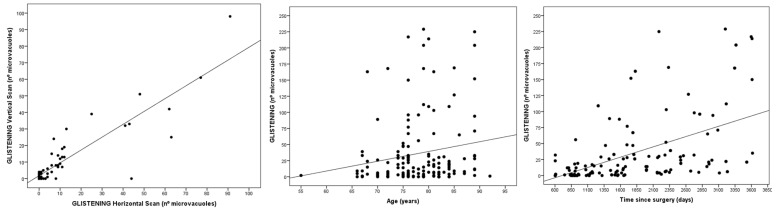
Correlation between horizontal and vertical glistening (**left**), severity of the glistening and age (**center**) and severity of the glistening and time since surgery (**right**).

**Table 1 jcm-12-02351-t001:** Demographics of the patients and intraocular lenses (IOL) included in the present study. Mean ± standard deviation (range).

Parameter	Value
Age (years)	70.2 ± 1 1.3 (55–92)
Sex (female/male, %)	54/46
Eye (right/left)	51.5/48.5
Time since surgery (days)	1843 ± 843 (603–3617)
IOL power (diopters)	20.3 ± 4.1 (8–27)
IOL model and material	SN60WF (acrylic hydrophobic)

**Table 2 jcm-12-02351-t002:** Reproducibility of OCT glistening measurements on the IOL (mean ± standard deviation) (N = 50 eyes).

Parameter		Values	Parameter	Values
**Intraobserver reproducibility**	**Measurement 1**	11.62 ± 20.01 (0–90)	**Severity scale 1**	0.82 ± 0.94 (0–3)
	**Measurement 2**	11.32 ± 19.30 (0–85)	**Severity scale 2**	0.82 ± 0.94 (0–3)
	**ICC**	0.994 (0.990–0.997)	**ICC**	0.977 (0.961–0.987)
**Interobserver reproducibility**	**Observer 1**	11.62 ± 20.01 (0–90)	**Severity scale observer 1**	0.82 ± 0.94 (0–3)
	**Observer 2**	11.54 ± 18.87 (0–87)	**Severity scale observer 2**	0.84 ± 0.98 (0–3)
	**ICC**	0.996 (0.993–0.998)	**ICC**	0.967 (0.943–0.981)
**Repeatability**	**Exploration 1**	11.10 ± 18.5 (0–82)	**Exploration 1**	0.78 ± 1.07
	**Exploration 2**	9.63 ± 17.11 (0–87)	**Exploration 2**	0.75 ± 1.06
	**ICC**	0.958 (0.927–0.975)	**ICC**	0.955 (0.926–0.973)

## Data Availability

Data used to support the findings presented in this study are available on request from the corresponding author.

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
