# Peer review of "Objective Classification of Glistening in Implanted Intraocular Lenses Using Optical Coherence Tomography: Proposal for a New Classification and Grading System"

_jcm, 2023, doi:10.3390/jcm12062351_

Round 1

Reviewer 1 Report

Fernandez-Vigo and colleagues report a novel analysis of IOL glistening using AS-OCT. The described technique is easy to reproduce and the classification scheme seems appropriate. I have the following comments:

- The authors repeatedly mention that this is the first study to utilize AS-OCT to identify IOL glistening. However, two studies which are not cited in this report have discussed a similar technique. Tripathy and Sridhar (Indian Journal of Ophthalmology, 2019) published an ophthalmic image of IOL glistening detected by AS-OCT, and Werner et al. (JCRS 2012) described various IOL changes -including glistening- in vitro, in a set of explanted IOLs and cadaveric eyes. While the report here remains novel in including a large cohort of in vivo analyzed IOLs and a proposed classification scheme, both mentioned reports should be cited and discussed.

- The reproducibility study is a bit confusing. The authors mention that they used a separate set of 50 eyes to asses interobserver agreement. Why not grade the study eyes by two independent graders to provide a more unbiased analysis of agreeability? Please elaborate on the methods of reproducibility analysis.

- I imagine that hyperreflective dots on the IOL can be anything, ranging from pigment deposits (which are common after phaco surgery) to exfoliative material to silicone oil..etc. How do the authors propose the distinction be made? Is concurrent slit lamp exam important to correlate with the AS-OCT image?

- In the introduction, the authors mention that there are 2 possible complications for IOLs (opacification and glistening). Complications of IOLs are much more numerous and the sentence should be rephrased.

Reviewer 2 Report

This manuscript proposed a new classification and grading system of IOL glistening based on SS-OCT examine and test its reproducibility.

Introduction

What does acronym “PMMA” mean? 

Why think OCT could be a helpful tool to evaluate IOL glistening? What is the evidence?

Methods

Were all the patients implanted with the same IOL?    (SN60WF) if so, is the conclusion of this study suitable for other IOLs?

Was the scale of Figure 2D different from Figure2A-C.

As mentioned in 2.3, the degree of glistening was classified into four categories based on a pilot study, is there any reference?

As a reader, I also want see the correlation between glistening severity and patients’ age or the time since surgery. 

Results

The result of the general study and correlation study could be summarized as a graph for a more visualized understanding.

The title of Table. 2, is that SS-OTCA? Or SS-OCT.

Discussion

Paragraph 3-7, this part of introducing several references were too long, and I can’t get the role of them. Maybe a brief summary of them, like what’s their shortcomings, could help to show the advantage of your study.

Paragraph 8-9 are more like the background of the IOL glistening. Besides, cut the length if still want it in the article.

Round 2

Reviewer 2 Report

The authors have addressed all my comments.